# Study protocol for economic evaluation of probiotic intervention for prevention of neonatal sepsis in 0–2-month old low-birth weight infants in India: the ProSPoNS trial

Anju Sinha [ID],[1] Pankaj Bahuguna [ID],[2,3] Subodh Sharan Gupta,[4] Yamini Priyanka Kuruba,[1] Ramesh Poluru [ID],[5] Apoorva Mathur,[1] Dilip Raja,[1] Abhishek V Raut,[6] Kamaleshwar S Mahajan,[6] Rishikesh Sudhakar,[5] Bharati Kulkarni [ID],[1] Ravindra Mohan Pandey,[7] Narendra K. Arora,[5] Shankar Prinja [ID] [2]

For numbered affiliations see end of article.

**Correspondence to**
Dr Anju Sinha;
apradhandr@gmail.com

## ABSTRACT

**Introduction** The ProSPoNS trial is a multicentre, double-blind, placebo-controlled trial to evaluate the role of probiotics in prevention of neonatal sepsis. The present protocol describes the data and methodology for the cost utility of the probiotic intervention alongside the controlled trial.

**Methods and analysis** A societal perspective will be adopted in the economic evaluation. Direct medical and non-medical costs associated with neonatal sepsis and its treatment would be ascertained in both the intervention and the control arm. Intervention costs will be facilitated through primary data collection and programme budgetary records. Treatment cost for neonatal sepsis and associated conditions will be accessed from Indian national costing database estimating healthcare system costs. A cost–utility design will be employed with outcome as incremental cost per disability-adjusted life year averted. Considering a time-horizon of 6 months, trial estimates will be extrapolated to model the cost and consequences among high-risk neonatal population in India. A discount rate of 3% will be used. Impact of uncertainties present in analysis will be addressed through both deterministic and probabilistic sensitivity analysis.

**Ethics and dissemination** Has been obtained from EC of the six participating sites (MGIMS Wardha, KEM Pune, JIPMER Puducherry, AIPH, Bhubaneswar, LHMC New Delhi, SMC Meerut) as well as from the ERC of LSTM, UK. A peer-reviewed article will be published after completion of the study. Findings will be disseminated to the community of the study sites, with academic bodies and policymakers.

**Registration** The protocol has been approved by the regulatory authority (Central Drugs Standards Control Organisation; CDSCO) in India (CT-NOC No. CT/NOC/17/2019 dated 1 March 2019). The ProSPoNS trial is registered at the Clinical Trial Registry of India (CTRI). Registered on 16 May 2019.

**Trial registration number** CTRI/2019/05/019197; Clinical Trial Registry.

## STRENGTHS AND LIMITATIONS OF THIS STUDY

⇒ This would be the first study evaluating cost-effectiveness of using probiotic intervention among low-birth weight newborn infants in India.
⇒ This study would inform incremental costs per disability-adjusted life year averted with use of probiotics, which will be key for prioritising resource allocation in view of other competing demands for child health, therefore crucial for public policy decisions.
⇒ Data from the RCT would be complimented with the use of a model-based approach to explore scenarios beyond the trial data.
⇒ Generalisability and external validity of findings from economic evaluation concurrently performed with the main study may not represent the same treatment effects and costs as in routine clinical practice.

## INTRODUCTION

There is progress towards attaining the third Sustainable Development Goal (SDG) of under-five (U5) mortality of 25 per 1000 live births by 2030, in India. Neonatal mortality rate is the most important contributing factor of the U5 mortality.[1 2] Further decline in neonatal mortality may require newer interventions for prevention or treatment of neonatal infections like meningitis, pneumonia, septicaemia that result in more than a quarter of the 1 million neonatal deaths every year in India.[2]

The ProSPoNS trial[3] proposes to evaluate the role of probiotics in prevention of neonatal infections (sepsis) among the vulnerable group of low-birth weight infants contributing the highest rates of morbidity and mortality. A larger sample size and

precise outcome measures differentiate this trial from the earlier conducted pilot study that showcased a 21% non-significant decline in morbidity due to neonatal sepsis in the intervention arm. Considering the 30% prevalence of low-birth weight (LBW) in India and 30% mortality due to neonatal sepsis, even a small decline in the incidence of sepsis would translate to a large number of new born lives saved.[3]

If the intervention is proven to be efficacious, it would be important to know whether it is also a cost-effective intervention. There are few examples in the literature where economic evaluations have been conducted on other effective interventions to prevent neonatal sepsis.[4 5] It cannot be overemphasised that along with clinical effectiveness of interventions, an estimate of economic effectiveness either in terms of costs or utilities (disability-adjusted life years; DALYs and Quality adjusted life years; QALYs) is required to make correct choices about their inclusion in public health programmes. This is of immense importance in presence of multiple competitive options where prioritisation is needed.

Ranjeva et al[6] have quantified the scope of the public health and economic burden of neonatal sepsis in Sub-Saharan Africa. However, evidence on the health economics aspect of neonatal sepsis prevention effect of probiotics has not been ascertained so far. We propose to evaluate the cost utility of probiotic intervention for prevention of neonatal sepsis within the ProSPoNS trial.

## BACKGROUND

ProSPoNS trial is a phase III multicentre randomised, double-blind placebo-controlled trial, being conducted at six study sites across India. It is a global Health trial funded by the UK Research and Innovation.

The Study is ongoing at six sites across India: Jawaharlal Institute of Postgraduate Medical Education and Research (JIPMER), Puducherry, Mahatma Gandhi Institute of Medical Sciences (MGIMS), Wardha, King Edward Memorial Hospital (KEM Hospital), Pune, Asian Institute of Public Health (AIPH), Bhubaneswar (with recruitments from SCB Medical College and Hospital, Cuttack), Subharti Medical College (SMC) and Lala Lajpat Rai Memorial Medical (LLRM) College District women hospital, Meerut and Lady Hardinge Medical College (LHMC), Kalawati Saran Children's Hospital (KSCH), Delhi. The primary objective is to estimate a 30% reduction in the incidence of neonatal sepsis in the intervention arm with daily supplementation of 1 mL (10 billion CFU/mL) of Vivomixx drops over a period of 30 days in 0–2-month-old LBW infants. 6144 low-birth weight (between 1500 g and 2500 g) infants would be recruited to the study from the six study sites located in Bhubaneswar, Meerut, New Delhi, Puducherry, Pune and Sewagram, between ages 3 and 7 days of life.

## Theory of change

More than one-third of the estimated four million neonatal deaths around the world each year are caused by severe infections, and a quarter—around one million deaths—is due to neonatal sepsis/pneumonia alone. Panigrahi et al in their study which was conducted in the year 2017 reported that the Culture-confirmed incidence of sepsis as 6.7/1000 births with 51% Gram negatives (Klebsiella predominating) and 26% Gram positives (mostly Staphylococcus aureus). The SDG for child survival cannot be achieved without substantial reductions in infection-specific neonatal mortality.[1] Due to non-specific signs and symptoms, sepsis is difficult to diagnose. The chances of recovery and survival depend on the correct diagnosis, referral, reaching a facility and adequate treatment by appropriately trained health workers and high-quality services.

Low-birth weight is one of the important indirect causes of deaths in neonates accounting for 40%–80% of neonatal deaths worldwide.[2] LBW neonates are immune compromised and manifest poor cognitive functions.[7] Since LBW infants are more prone to infections, neonatal mortality and morbidity may be decreased by preventing neonatal infection.

Antibiotic therapy is the mainstay of current therapeutic management of neonatal infections. For serious infections in new borns and young infants, there is a lack of guidelines on rational antibiotic use.[8] In the absence of rapid diagnostics, non-judicious use of antibiotics contributes to the rising antimicrobial resistance. Antibiotic overuse is linked with disturbing the gut ecosystem. The problem of drug resistance outweighs the fast pace of newer generation antibiotic production.

Currently, there are no preventive interventions available for neonatal sepsis other than general measures of hand washing, exclusive breastfeeding, etc. However evaluating immunotherapy with immune globulin, probiotics, myeloid colony-stimulating factors, glutamine supplementation, recombinant human protein C and lactoferrin are being evaluated as adjuvants for the prevention of neonatal sepsis.[9] Probiotics are non-pathogenic microbes which are capable of exerting positive health effects in the host body through various mechanisms.[10 11] It acts by suppressing the growth of potential pathogens and their epithelial attachment. It also works by secreting antimicrobial substances or by stimulating host expression of protective molecules. Besides, it can produce a barrier against the harmful pathogens at higher levels. It stimulates the process of immunosuppressive agent's production in the host which suppress the inflammatory responses or accelerate the immunologic mechanism that clears the infection. Probiotics stimulate normal microbe–microbe and host–microbe interactions positively, thereby enhance the protection provided by commensal flora through competitive interactions, pathogenic antagonism and antimicrobial factors. We hypothesise that immunomodulation/immunopotentiation with probiotics may prove to be an alternative for the prevention of neonatal sepsis.

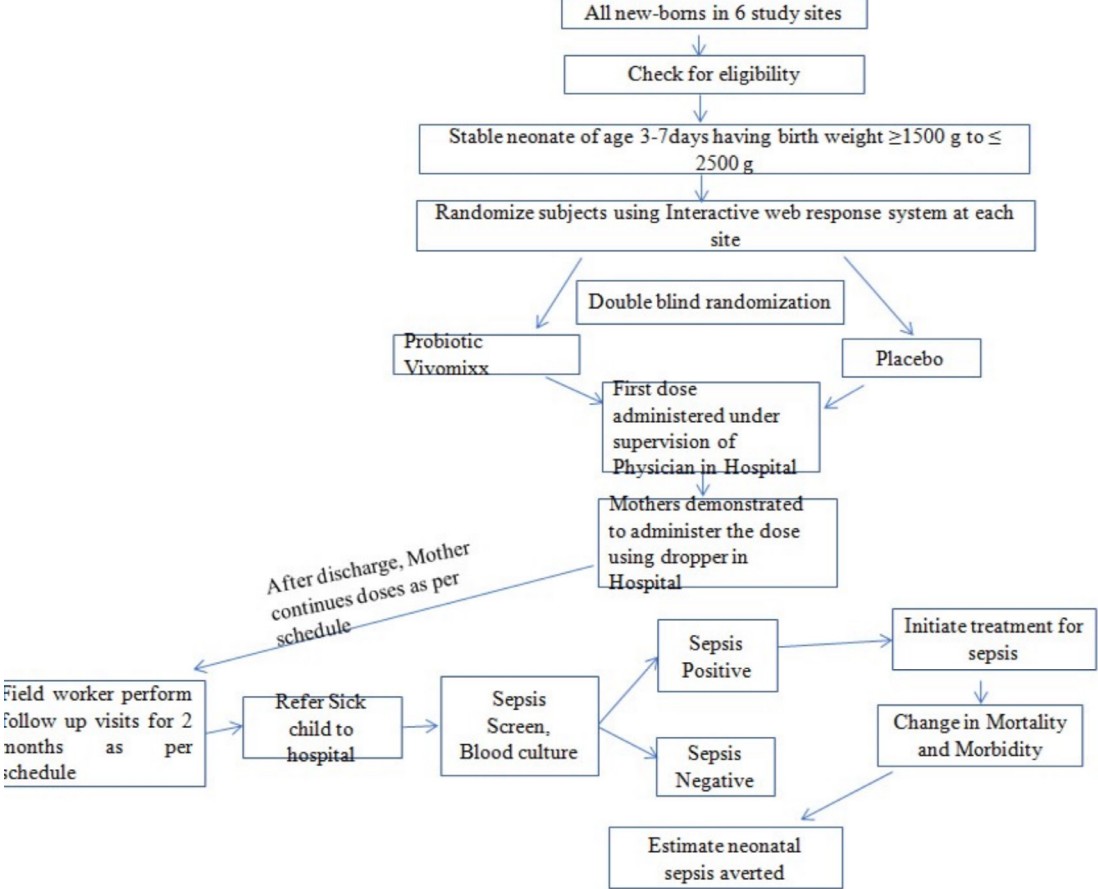

**Figure 1** Conceptual framework to prevent neonatal sepsis using probiotics.

Figure 1 depicts the conceptual framework to prevent sepsis episodes among 0–2 months neonatal population with use of probiotics.

### Research question

Is probiotic (Vivomax) use as prophylaxis a cost-effective strategy for preventing sepsis among 0–2-month-old infant population in India?

Objectives: to assess the cost utility of using probiotics as prophylaxis for sepsis among LBW neonates of age 0–2 months.

### Study methodology

A cost utility analysis (CUA) is proposed to be conducted alongside the multicentre RCT. Randomised controlled trials (RCT) are considered the 'gold standard' as they produce a reliable the best evidence of treatment effect. The economic evaluation alongside RCT provides opportunity to collect data both on disease-specific disability and resource use. Start date is 15 November 2022 and the anticipated completion date is December 2023.

### Model overview
#### General description

This protocol describes the methodology for economic evaluation within a phase III RCT—the ProSPoNS trial. The details are described in the following sections.

#### Decision model

In view of the shortcomings of treatment effect measure coming from a single RCT, we also propose using a model-based approach which will allow synthesising information from all sources for a health technology enabling us to explore scenarios not explicitly found in the trial data. Moreover, the modelling approach would enable to estimate the long-term benefits of intervention incorporating quality of life measures. Furthermore, it allows accounting for uncertainties present in assumptions and data used for economic evaluation, which has implications for policy question under consideration. The clinical outcomes from trial would be sepsis episodes averted as an effect of probiotics use in the intervention arm. Quality of life (QUALY) is a preferred utility measure for economic analysis; however, there is no EQ-5D version for neonates. Therefore, in order to assess the long-term effects, we propose to use 'DALY' as the measure of health outcome, which accounts for both disability and quality of life. A decision-analytic model following a systematic approach in bringing together evidence on clinical effectiveness, health-related disability of life and costs associated with the probiotic intervention as compared with placebo will be used with the aim to provide information on whether the reduction in neonatal sepsis incidence justifies the incremental costs. Figure 2 depicts the decision model for the proposed study.

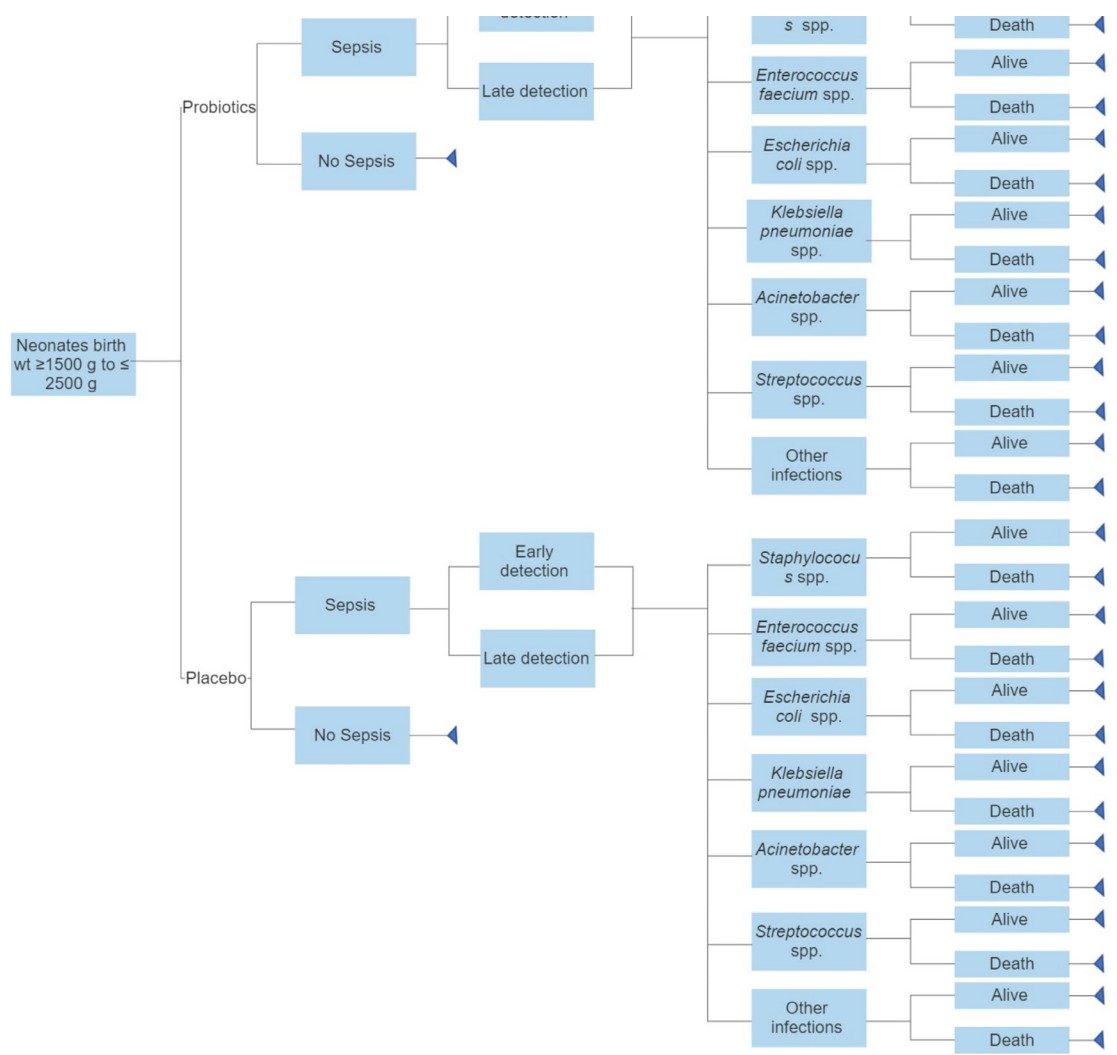

**Figure 2** Decision tree for cost-effectiveness of using probiotics as prophylaxis for sepsis among neonates of age 0 to 2 months.

## Outcome measures

Primary outcome measure for economic evaluation (within-trial CUA) would be sepsis episodes and related deaths averted in the intervention arm as compared with the placebo arm, which translates to gain in life years and DALYs averted as long-term measure.

| Table 1 Data requirements for economic evaluation | | |
|---|---|---|
| **Data type** | **Significance for the model** | **Source of data** |
| Demographic | ▶ Estimate the baseline parameters<br>▶ Estimate target population | Primary data collection; Census of India |
| Treatment effects and incidence of adverse events | ▶ Efficacy and safety parameters<br>▶ Cost estimation<br>▶ Outcome assessment | RCT Database |
| Background mortality data | Estimate natural mortality in the target population | SRS, Published literature |
| Utilities | Estimate DALY as long-term outcome measure | Global Burden of Diseases estimates (2011) |
| Resource use and price data | Cost estimation | Primary data collection; NHSC—Prinja *et al* |
| Epidemiological data | Estimate baseline risk of clinical events | Published literature; Primary data collection (Control arm of the RCT) |
| NHSCD, National Health Systems Costing Database; RCT, randomised control trial; SRS, Sample Registration System. | | |

Data requirements: data requirements for economic evaluation are tabulated in table 1.

Utility data would allow incorporating the relative impact of a health technology (ie, probiotics) on the neonatal health outcomes in terms of disability burden. This will be estimated as DALYs averted. Utility data would be obtained from Global disease burden estimates 2017. Cost data are important drivers of cost-effectiveness, as they quantify the resource consumption associated with the use of health technologies.

Decision-analytic model would be populated with cost data from the societal perspective: direct medical and non-medical cost borne by patients; and direct medical cost borne by the healthcare system. A National Health Systems Costing Database (NHSCD) to estimate healthcare system's costs in India has been developed.[12] Data on patient's out-of-pocket (OOP) expenses would be collected during the conduct of the RCT as primary data.

A health economic evaluation using CUA design will be conducted within the ProSPoNS trial. The results would provide the incremental costs and health gains of the probiotic intervention in absence of any prevention modality being implemented or available. Decision-analytical modelling using the trial results would be extrapolated to the target population and time horizon to inform policy on the potential for incremental costs offsets attributable to prevention of morbidity with the proposed intervention. The CUA will use the present results as cost per case of neonatal sepsis prevented. CUA will present the results as cost per DALY averted. Intervention costs and effects will be extrapolated to the Indian neonatal population. Modelling will be used for the target population, time horizon and the decision context to provide much required information for the policymakers. It would guide further decision for upscaling the intervention, if it is found to be efficacious. If the incremental costs of the intervention can be offset by the gains in terms of effectiveness (ie, DALYs), it may be presented for induction within the ambit of existing governmental programme. Modelled CUA results can be presented as health-adjusted life years saved and savings of the healthcare system by disease prevention. We will calculate the incremental cost-effectiveness ratios as the difference in the costs between the intervention and the comparator divided by the difference in health outcomes.

## Effectiveness assessment

Prevention of severe sepsis in the early neonates will prevent the long-term morbidity like neurodevelopmental disability risk associated with sepsis[13] and reduce mortality due to pneumonia, meningitis and neonatal sepsis and hospital admissions due to any infectious diseases.[14]

We propose to use the short time horizon to assess the sepsis averted as a result of probiotics supplementation for 1 month and model the change in DALYs in the intervention group as a long-term measure.

## Study details

Total of 6144 (N) new born babies with LBW would be screened for eligibility and randomly assigned to probiotic (n=3072) or placebo (n=3072) groups and administered 1 mL of the probiotic suspension daily for 30 days. The study would use Interactive Web Respons System (IWRS) for the purpose of stratified randomisation by birth weight, sex and study site. Trained field workers will make home visits daily for first 7 days and thereafter three times in a week till day 30 of the follow-up period to ensure and monitor adherence to the intervention and collect morbidity information. After the 30-day intervention period, home visitation will continue at weekly interval till day 60 of life. During home visitation as per schedule, trained fieldworkers would screen infants for symptoms/signs of possible serious bacterial infection (PSBI)/clinical illness/clinical severe infection. To record the adverse event (AE)/serious adverse event (SAE) (if any), continuous monitoring and reporting of ongoing medical conditions, signs and symptoms will be conducted by field worker during their scheduled follow-up home visits using software on handheld device during the 2-month period of follow-up. Presence of any danger signs would be evaluated, and need for referral would be assessed: if a child is found to have any symptoms/signs from the list, the field worker will accompany the infant to facility for examination by study physician (referral). The study physician would examine the infant again for symptoms/signs of PSBI/clinical illness/clinical severe infection and confirm the diagnosis. Investigations such as blood culture, sepsis screen would be performed for suspected cases of neonatal sepsis by study physician. If required, the infant would be hospitalised for treatment. Data on outcomes, lab investigation and treatment during clinical management of the sick, new born will be collected and SAEs will be reported as per regulatory guidelines. Blood culture isolates will be shipped by site to microbiology lab at All India Institute of Medical Sciences, New Delhi for quality assurance purpose.

The study will be implemented according to MRC guidelines for management of global health trials and will be governed by a Trial Steering Committee and a Data Management Committee.

## Primary outcomes

1. *1. Sepsis:* defined as one or more clinical symptoms suggestive of sepsis with a microbial isolate on blood culture; or a neonate with sterile blood culture with at least 2 sepsis screen markers being abnormal (absolute neutrophil count <1500/mm$^3$; Total Leucocyte Count (TLC) <5000/mm$^3$; immature to total neutrophil ratio >0.2; immature to total neutrophil ratio >0.2; C reactive protein (CRP) >1.2 mg/dL, ESR >15 mm).

2. *2. PSBI:* one or more clinical signs like fever (temperature ≥38°C), low body temperature (< 35.5°C), not feeding well, convulsions, fast breathing (60 bpm or more), severe chest in-drawing, movement only when

stimulated or no movement at all in infants less than 7 days old confirmed by study physician.

### Secondary outcomes

1. *1. Clinical severe infection:* one or more clinical signs like severe chest in-drawing, not feeding well, fever (temperature ≥38°C), the movement only when stimulated as confirmed by the study physician, low body temperature (< 35.5°C).

2. *2. Critical illness:* one or more of clinical signs like unable to feed at all, unable to cry, no movement on stimulation, bulging fontanelle, cyanosis as confirmed by the study physician and convulsions. The flowchart below (figure 1) depicts the study design and the study processes that will be followed during the implementation of the proposed research work (diagram, to be finalised after finalisation of methods). Primary and secondary outcomes would be compared between the two study arms to assess clinical effectiveness of probiotic intervention.

### Costing

In the current study, cost of Vivomix (intervention), cost of routine home-based postnatal care, cost of illness management, drug costs, vitamin/mineral supplement costs, laboratory investigation charges, doctor consultation and bed charges are categorised as direct medical costs and the informal payments and travel are categorised as the direct non-medical costs. Loss of wages of parents and other caregivers due to absenteeism from work and is categorised as indirect costs. Neonatal sepsis episodes are treated in in-patient (IP) setting; therefore, respondents (parents and caregivers) will be interviewed to collect information on days they were absent from work due to hospital stay for treatment. After hospital discharge, follow-up with parents and caregivers to assess any effect on their productivity at work is beyond the scope of data collection in the present study. All costs related data would be collected from all six study sites in a representative manner.

*Sample size (costing):* we calculated the sample size to estimate the mean OOP expenditures among the neonates with illness (here PSBI). A power of 80% and 5% level of significance was assumed for sample size calculation. The baseline estimate for OOP among neonates with PSBI was accessed from an Indian study.[1] Assuming a mean OOP of ₹2003 (SD: ₹3517) for neonates with PSBI and an error margin of 7.5%, the minimum sample size computed to estimate the OOP was 1998. In view of high variability in the OOP data, the error margin of 7.5% was considered appropriate for sample size calculation. The data on OOP will be captured for a subsample, that is, 1998 neonates, out of total sample recruited in the study. At each of the six subjects' recruitment sites, data would be collected on 333 subjects for this substudy. The OOP data will be collected irrespective of status of neonates belonging to two study arms. The formula used for sample size calculation is given below: n=$\frac{Z2_{1-a/2}}{}$s2 d$^2$

$$n = \frac{Z^2_{1-a}/2S^2}{d^2}$$

Where
n=minimum estimated sample size.
$Z^2_{1-a}$ = table value from standard normal distribution.
$s^2$=SD of outcome variable in study population.
$d^2$ = error margin (or absolute precision).

*Data collection and estimation of costs:* data collection for the current study would be completed in 1 year duration from September 2022 to August 2023 (to cover any variation with seasonality). Data on OOP expenditures will be collected on a subsample of 333 subjects drawn from the total trial population in a representative manner.

Sociodemographic data on education, occupation of parents of enrolled children is being collected on study forms. Parents would be asked about any illness during the study period and any expenditures on account of care seeking, visits to facilities, hospitalisation and treatment. Direct medical and non-medical costs due to neonatal sepsis would be ascertained in all 333 subjects at the six study sites. Data on OOP expenditure in public or private health facilities would be elicited.

Intervention cost would be facilitated through primary data collection and the programme budgetary records. The cost of treatment for neonatal sepsis and related diseases at secondary and tertiary level will be ascertained from the NHSCD[15 16] and published literature in Indian context.[12] These sources provide estimates on cost per IP admission, cost of intensive care unit admission, cost per out-patient consultation accounting for resources consumed for service provision such as medical (drugs, diagnostics, procedural, hospital charges) and non-medical (transportation, boarding and lodging). In addition, cost related to Information Education and Communication (IEC) material, monitoring and supervision and other administrative costs will also be estimated. Assumptions made in the study protocol are provided in table 2.

Total cost will be comprised of cost of intervention, cost of management of sepsis and associated complications and patient costs as OOP. Similarly, total cost in the control arm will be estimated in the absence of intervention. Annual discount rate of 3% will be used wherever applicable. In the present study, both model-based approach as well as trial data would be used. Data management and analysis will be done using MS-Excel.

### Other analyses

Uncertainty analyses, budget impact analysis, stakeholder analysis would be included.

### Uncertainty analysis

Both deterministic and probabilistic sensitivity analysis (PSA) will be undertaken to address the effect of uncertainties present in analysis. For PSA, all the parameter values will be varied within a plausible range. Based on nature of parameter, an appropriate probabilistic distribution will be assigned such as beta distribution for risk and

**Table 2** Key methods and assumptions for economic evaluation

| Component | Consideration/approach | Justification |
|---|---|---|
| Intervention | ► Vivomixx (eight strain probiotics) + sepsis management<br>► Dose: 1 mL/day (10 billion CFU/mL) for 30 days<br>► Intervention delivery: under the supervision of study staff (field worker) | Effectiveness of Vivomixx for preventing neonatal sepsis is being assessed under trial |
| Comparator | Placebo | ► Effectiveness of Vivomixx is being assessed under trial against placebo<br>► Use of antibiotic is routine practice for sepsis management. |
| Target population | High-risk LBW neonates (both pre-term and small for-date babies) | Most vulnerable high-risk population with high morbidity and mortality |
| Objective | To assess the incremental cost per DALY averted with use of probiotics for prevention of neonatal sepsis compared the placebo | Besides effectiveness, evidence on cost-effectiveness is mandatory for adoption of treatment/intervention under public policy |
| Target Audience/policy use | ► National Health Mission (NHM)<br>► Any other | This intervention is to be implemented under RMNCH programme |
| Method | Both model and trial based | ► Short –term outcome estimates will be used for trial<br>► Use of model-based CEA will enable us to assess long-term outcomes (and sub-group analysis) |
| Perspective | Societal perspective | OOP burden associated with sepsis management among LBW neonates |
| Analytic horizon | 6 months | To model long-term outcomes (DALYs) |
| Costs | ► Health system costs (Vivomixx cost; Cost of routine HBPNC care; cost of illness management etc.)<br>► OOP expenditures (doctor's consultation; bed charges; drug costs; vitamin/mineral supplement costs; investigations; procedures; informal payments; travel cost etc. | Both health system and patients' costs as per societal perspective considered for CEA |
| Outcomes | ► Sepsis episodes and deaths averted<br>► DALYs<br>► Hospitalisation averted<br>► Life years gained | In line with study design (cost-effectiveness/cost-utility analysis) |
| Discount rate | 3% | As per Indian reference case |

CEA, Cost Effectiveness Analysis; DALYs, disability-adjusted life years; HBPNC, home-based postnatal care; LBW, low-birth weight; OOP, out-of-pocket.

utility parameters, gamma distribution for cost parameters, uniform distribution for demographic parameters, etc. The results under PSA will be simulated 999 times to report a 95% CI for ICERs.

### Budget impact analysis

A budget impact analysis is proposed to be conducted to estimate the financial consequences of increasing uptake of probiotic supplementation for LBW infants in the neonatal care under NHM, over the near future (3 to 5 years). Data regarding the target population of LBW infants will be available by extrapolating the data from the RCT, as will be other epidemiological inputs to populate the budget-impact model. At present, there are no alternative prevention modalities for neonatal sepsis present (other than hand hygiene and exclusive breastfeeding). Therefore, assumption would be made regarding the uptake of probiotic supplementation (based on the results of the study).

### Stakeholder analysis

A range of potential stakeholders will be consulted, involving experts such as neonatologists/paediatricians, academics, policymakers, programme managers, industry, patient group representatives, etc. Multiple consultations

with stakeholders will be done during the conduct of study as well as dissemination of study results.

## PATIENT AND PUBLIC INVOLVEMENT

Patients or caregivers or members of public were not involved in the development of this protocol.

## DISCUSSION

The ProSPoNS trial is evaluating the role of probiotics intervention in prevention of neonatal infections, a major cause for neonatal mortality. To date, no prevention modalities other than hand hygiene and exclusive breast feeding have been successfully incorporated in programmes for prevention of neonatal infections. Panigrahi *et al* established benefit of *Lactobacillus planitarum* with fructooligosaccharides intervention in prevention of sepsis and related deaths in new borns; however, the study did not assess the cost-effectiveness of the intervention. There is no study in the literature evaluating the cost-effectiveness of this novel disease prevention technology. We have proposed such evaluation alongside the ProSPoNS trial.

An RCT design, considered a gold standard for such evaluations, is being used in the trial. However, there are disadvantages of using RCT data quoted in the literature. Data from a single RCT may not be adequate for estimation of cost utility as RCTs mainly focus on a single intervention and have small follow-up period, thus, lack to provide evidence on effectiveness of multiple alternatives and situations beyond the trial conditions. The treatment effect measured in a single study may be relevant for a particular setting but has limited generalisability of results in populations other than the one studied.[17–19] We propose using a modelling approach to overcome the above-mentioned shortcomings of RCT data.

There are a few examples of estimation of cost-effectiveness of disease preventive technologies in the neonatal population. A study conducted in Sub-Saharan Africa estimated that neonatal sepsis resulted in 5.29–8.73 DALYs lost annually accounting for an economic burden from \$10 billion to \$469 billion (per annum). It was recommended that a strategic plan for the treatment and prevention of neonatal sepsis may reduce this burden.[6] A cost-effective analysis conducted in Ghana reported that alcohol-based hand rub (ABH) effectively reduced the patient cost of neonatal bloodstream infection (BSI), hospital cost associated with BSI, length of hospital stay and deaths due to BSI by 41.7%, 48.5%, 50% and 73%, respectively. It was also estimated that ABH reduced the incidence of hospital-acquired neonatal infections by 16%.[4] Current study would estimate the cost-effectiveness of a novel neonatal sepsis prevention modality.

In considering cost-effectiveness, measures on treatment effect over longer time horizon than RCTs are often required. Therefore, we have proposed a modelling approach in addition to the use of primary data coming from the ongoing RCT. This approach would use and provide a wider range of assumptions/parameters from multiple data sources and evaluate the long-term utility of the probiotic intervention in disease prevention.

Expected outcomes of study and policy implications: the health economic evaluation would generate data on cost involved with the new intervention as compared with the control arm with uncertainty range, incremental costs and health outcomes for both study arms with uncertainty range, ICER, sensitivity and threshold analyses. It would be helpful for the policymaker to understand the cost utility of the probiotic intervention in the presence of other competing priorities.

### Strengths

► This would be the first study evaluating cost-effectiveness of using probiotic intervention among low-birth weight newborn infants in India.
► This study would inform incremental costs per DALY averted with use of probiotics, which will be key for prioritising resource allocation in view of other competing demands for child health; therefore, crucial for public policy decisions.
► Data from the RCT would be complimented with the use of a model-based approach to explore scenarios beyond the trial data.

### Challenges

► Generalisability and external validity of findings from economic evaluation concurrently performed with the main study may not represent the same treatment effects and costs as in routine clinical practice.

### Registration

The protocol has been approved by the regulatory authority (Central Drugs Standards Control Organisation; CDSCO) in India (CT-NOC No. CT/NOC/17/2019 dated 01 March 2019). The ProSPoNS trial is registered at the Clinical Trial Registry of India (CTRI). Registered on 16 May 2019

### Ethics and dissemination

Approval has been obtained from EC of each of the six participating sites (MGIMS Wardha, KEM Pune, JIPMER Puducherry, AIPH, Bhubaneswar, LHMC New Delhi, SMC Meerut) as well as from the ERC of LSTM, UK. A peer-reviewed article will be published after completion of the study. Findings will be disseminated in the community of the study sites, with academic bodies and policymakers.

**Author affiliations**
[1]Reproductive, Child Health and Nutrition, Indian Council of Medical Research, New Delhi, Delhi, India
[2]Department of Community Medicine and School of Public Health, Post Graduate Institute of Medical Education and Research School of Public Health, Chandigarh, India
[3]School of Health and Wellbeing, College of Medical, Veterinary and Life Sciences, Health Economics and Health Technology Assessment, University of Glasgow, Glasgow, Scotland's Western Lowlands, UK
[4]Community Medicine, MGIMS, Wardha, India

[5]Research Department, The INCLEN Trust International, New Delhi, India
[6]Department of Community Medicine, Mahatma Gandhi Institute of Medical Sciences, Sevagram, India
[7]Department of Biostatistics, All India Institute of Medical Sciences, New Delhi, India

**Contributors** AS, SSG, NKA, AVR, RP conceptualised and developed the research question. PB, SP, YPK, AM, DR developed the methodology, tools for data collections and framed the analysis plan. BK, SSG, AS, KSM, RS, RMP, NKA involved in study protocol writing, review and editing. AS, PB, YPK developed the original draft of manuscript.

**Funding** The study is funded by the Department of Health and Social Care (DHSC), the Foreign, Commonwealth & Development Office (FCDO), the Medical Research Council (MRC) and Wellcome. Award/grant number is MR/S004912/1. The funding body has no role in the designing of the study and collection, analysis, and interpretation of data and in writing the manuscript.

**Competing interests** None declared.

**Patient and public involvement** Patients and/or the public were not involved in the design, or conduct, or reporting, or dissemination plans of this research.

**Patient consent for publication** Consent obtained from parent(s)/guardian(s).

**Provenance and peer review** Not commissioned; externally peer reviewed.

**ORCID iDs**
Anju Sinha http://orcid.org/0000-0002-0830-7723
Pankaj Bahuguna http://orcid.org/0000-0001-9952-6077
Ramesh Poluru http://orcid.org/0000-0002-7693-418X
Bharati Kulkarni http://orcid.org/0000-0003-0636-318X
Shankar Prinja http://orcid.org/0000-0001-7719-6986

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
