## [Reviewer comments · BMJ Open]

ARTICLE DETAILS

TITLE (PROVISIONAL)	A Study Protocol for Economic Evaluation of Probiotic Intervention for Prevention of Neonatal Sepsis In 0-2 Month Old Low Birth Weight Infants In India - The ProSPoNS Trial
AUTHORS	Sinha, Anju; Bahuguna, Pankaj; Gupta, Subodh; Kuruba, Yamini Priyanka; Poluru, Ramesh; Mathur, Apoorva; Raja, Dilip; Raut, Abhishek; Mahajan, Kamaleshwar; Sudhakar, Rishikesh; Kulkarni, Bharati; Pandey, Ravindra Mohan; Arora, Narendra K.; Prinja, Shankar

VERSION 1 – REVIEW

REVIEWER	Singh, Balpreet Dalhousie University
REVIEW RETURNED	20-Nov-2022

GENERAL COMMENTS	Thank you for the opportunity to review the study protocol on this crucial topic. This study plans to do an economic evaluation of probiotic supplementation in low-birth-weight infants for the prevention of neonatal sepsis. The cost-utility analysis will be undertaken using a societal perspective. Both prospective analyses as part of RCT for short-term outcomes and a decision model-based approach to assess long-term benefits will be undertaken. Overall, the economic evaluation part of the study is well described. Please see my comments below: 1. Please provide more details about the probiotic species being used in the preparation. Is the selected probiotic species shown to be effective in previous trials? Also, how will it be ensured that the probiotic preparation is being stored in appropriate conditions at home?2. The definition for sepsis (Primary Outcome) includes culture-proven sepsis, culture-negative sepsis with positive lab markers and clinical sepsis. The sensitivity and specificity of only lab markers or clinical sepsis are low and can result in significant variability in the incidence of the primary outcome. Why not limit the primary outcome to only culture-proven sepsis? Also, are positive urine or CSF cultures included in the sepsis definition?3. Will the severity of sepsis episodes, e.g. meningitis and need for intensive care, also be considered during the economic evaluation? Will other outcomes e.g. rate of necrotizing enterocolitis also taken into consideration during evaluation?4. In terms of the study population, why were babies <1500 grams excluded from the study? What are other exclusion criteria? Are babies with early onset sepsis excluded from the study?
---

	5. Please provide details regarding the consent process in the protocol. 6. Will indirect costs related to time loss from work be included in the decision model? 7. The cost of public and private healthcare systems varies significantly. How will that be taken into account during the economic evaluation? 8. Is there a plan to do an a priori subgroup analysis? e.g. based on gestational age? 9. How will other co-interventions, e.g. hand hygiene and breastfeeding rates, be compared between the two groups, and is there any plan to adjust for it in the analysis?
--	---

REVIEWER	Otieku, Evans University of Ghana, Institute of Statistical, Social and Economic Research
REVIEW RETURNED	15-Dec-2022

GENERAL COMMENTS	It is good to attempt trying to publish this protocol for an ongoing economic evaluation study. However, it did not seem that a health economist was part of the protocol design. Some clarity is needed to improve the quality of the protocol.  1. The study has no clear definition of what neonate means. If it applies to newborns aged <28 days old (<4 weeks) at the time of sampling, how is the 0 to 2 months old fit the definition of neonates? 2. There is no clear statement of the research question. 3. The authors intend to use DALY averted as a utility measure. However, DALY weights are predetermined and obtainable from the global burden of disease reports and are not suitable for clinical trial cost-effectiveness analysis. Instead, QALY gained with the intervention is a preferred utility measure. 4. It is unclear whether the protocol reporting followed specific guidelines. Many sections were misplaced. For example, Figure 1 appeared on page 4 but is referred to on page 9, line 40, as below. It will be best to merge the section on Costing and Estimation of Costs under one heading and include another section on the Measurement of Effectiveness. 5. Spell out all abbreviations in full at the first instance. For example, EC, ERC, MRC, IWRS, PSBI, AE/SAE, etc. 6. Study Trial: Page 8, line 52, presence of any danger sign....." what is this danger sign. 7. Specifically, what is the difference between the primary and secondary outcomes? 8. How do they intend to characterize heterogeneity and distributional effects if they collect data from six different geographic locations? 9. Why didn't they also consider the cost of productivity loss due to presenteeism? 10. In-text citation styles are not consistent. Some come after a period. Some are enclosed in (), while others are in [], making it difficult to read as values or references. 11. The authors said all neonates will be followed up at home for the first seven days. Are they not anticipating some neonates will be at the hospital due to their low birth weight? Does it mean if a neonate is still on admission they will be excluded? What exactly is their exclusion criteria? 12. They should rephrase the following for clarity:
--

	Page 3, lines 19 to 25, and lines 35 to 39. They should delete lines 46 to 48 (repetition).
REVIEWER	Purkayastha, Jayashree Kasturba Medical College Hospital, Paediatrics
REVIEW RETURNED	16-Dec-2022
GENERAL COMMENTS	Whether the babies included in the study are preterm or term, SGA, AGA is not mentioned. Only LBW is mentioned which has to be clearly defined. If preterm babies included then they will have prolonged NICU stay, which day of life you will start the probiotic is not defined. Please mention the cut off gestational age and birth weight. PSBI and critical illness are overlapping. Why did you choose Vivomixx as the probiotic in the study should be clarified.

VERSION 1 – AUTHOR RESPONSE

Reviewer: 1

Dr. Balpreet Singh, Dalhousie University

Comments to the Author:

Thank you for the opportunity to review the study protocol on this crucial topic. This study plans to do an economic evaluation of probiotic supplementation in low-birth-weight infants for the prevention of neonatal sepsis. The cost-utility analysis will be undertaken using a societal perspective. Both prospective analyses as part of RCT for short-term outcomes and a decision model-based approach to assess long-term benefits will be undertaken.

1 Sinha AP, Gupta SS, Poluru R, Raut AV, Arora NK, Pandey RM, Sahu AR, Bethou A, Sazawal S, Parida S, Bavdekar A. Evaluating the efficacy of a multistrain probiotic supplementation for prevention of neonatal sepsis in 0–2-month-old low birth weight infants in India—the “ProSPoNS” Study protocol for a phase III, multicentric, randomized, double-blind, placebo-controlled trial. *Trials*. 2021 Dec;22(1):1-2.

Overall, the economic evaluation part of the study is well described. Please see my comments below:

1. Please provide more details about the probiotic species being used in the preparation. Is the selected probiotic species shown to be effective in previous trials? Also, how will it be ensured that the probiotic preparation is being stored in appropriate conditions at home?

Response: The probiotic preparation Vivomixx is a mixture of eight strains: *Streptococcus thermophilus*, *Bifidobacterium breve*, *Bifidobacterium longum*, *Bifidobacterium infantis*, *Lactobacillus acidophilus*, *Lactobacillus plantarum*, *Lactobacillus paracasei* and *Lactobacillus delbrueckii* spp *bulgaricus*, at a dose of 10 billion cfu.

The results from a pilot study entitled “Effect of Probiotics VSL#3(Vivomixx) on prevention of sepsis in LBW infants during the 0-2-month period: A Randomized Controlled Trial” conducted by the Indian Council of Medical Research provides an indication that microbial interference by beneficial bacteria is helpful in decreasing neonatal morbidity (15). The supplementation with the probiotic VSL#3 (Vivomixx) in LBW infants was associated with a 21% reduction in the risk of suspected sepsis (PSBI) diagnosed by the field worker. In the post-hoc analyses using the physician’s diagnosis of sepsis as the outcome measure there was a 33% overall reduction in the risk of sepsis.

The Investigational product is stored at home under cold chain (2-8 degree Celsius) in the refrigerator or in a vaccine/day carrier. The ice packs in the vaccine carriers are replaced at

regular intervals by the field investigators during field visits. (PI refer to the published protocol of the trial: TRIALS 2021)

2. The definition for sepsis (Primary Outcome) includes culture-proven sepsis, culture-negative sepsis with positive lab markers and clinical sepsis. The sensitivity and specificity of only lab markers or clinical sepsis are low and can result in significant variability in the incidence of the primary outcome. Why not limit the primary outcome to only culture-proven sepsis? Also, are positive urine or CSF cultures included in the sepsis definition?

Response: Thank you for the comment. The study was initiated after the scrutiny of the protocol and approval by the ICMR Technical Advisory Group and UKRI MRC expert panel. The clinical study is already initiated and we have reached half of the enrolments now. We understand your concern regarding sensitivity and specificity. The analysis would be done for each of the parameters separately and in combination and presented. The rate of culture positivity in blood samples is quite low and hence the choice of sepsis screen was made as an alternative. Moreover, we anticipated very few cases of culture proven sepsis considering total referrals, and refusals for blood collection in the community setting. Therefore, we couldn't limit the primary outcome to only culture-proven sepsis. As the protocol is already published (Sinha et al TRIALS 2021) and being implemented, changes in the outcome assessment at this stage cannot be done. Urine/CSF culture are not mandatory for the diagnosis of sepsis in this study, however, these tests would be done depending upon clinicians' judgment, and as per the clinical situation.

3. Will the severity of sepsis episodes, e.g., meningitis and need for intensive care, also

be considered during the economic evaluation? Will other outcomes e.g., rate of necrotizing enterocolitis also taken into consideration during evaluation?

Response: All illnesses leading to hospitalisation/prolonged hospitalisation/ medical intervention will be considered during evaluation. Necrotizing enterocolitis will also be considered whenever reported by treating clinicians.

4. In terms of the study population, why were babies <1500 grams excluded from the study? What are other exclusion criteria? Are babies with early onset sepsis excluded from the study?

Response: Generally, the babies <1500grams are not compatible enough to accept oral feeding. The probiotics intervention is administered orally therefore the babies with birth weight <1500grams were excluded from study. As per the protocol, the babies who are in Stable clinical condition as assessed by a physician and accepting feeds orally (where stable is defined as, does not require intravenous fluids and vasopressor medication to maintain circulation and accepts oral feeding /breastfeeding) are included in the study and the New-born with illness requiring prolonged hospitalization and interference with oral feeding are excluded.

The exclusion criteria of study are listed below:

1. New-born with extreme prematurity (<34 weeks),
2. New-born with illness requiring prolonged hospitalization and interference with oral feeding
3. Presence of a gross congenital malformation incompatible with life
4. Parent/ Legal authorized representative (LAR) not providing written consent
5. Please provide details regarding the consent process in the protocol.

Response: We are taking separate written consent from the parents of the baby for this health economics sub study. The consent form is explained to the parents in their vernacular language. The parents are informed that their participation is voluntary. The information that will be collected from parents for this sub study is also explained to them. If the parents have any queries, then the queries are answered by study personnel. The understanding of the

parents about the study and the consent form is assessed by study personnel by asking them few questions regarding the study. If they agree to participate in the study, they are requested to sign the consent form. After parents signature the consent form is signed by study personnel and a copy of the signed consent form is provided to the parents.

6. Will indirect costs related to time loss from work be included in the decision model?

Response: Thanks for seeking the clarification regarding the indirect costs. We would like to clarify, indirect costs in terms of wage loss of parents and other caregivers will be captured as a part of primary data collection (given on page 10, lines 32-33). This will be considered in decision model while estimating the cost associated with two study arms. Though we do not consider any long-term productivity losses among parents of enrolled infants as a result of major disabilities such as neurodevelopmental risks due to sepsis. Moreover, it is obscure to do such estimations and may be less relevant from programmatic perspective.

7. The cost of public and private healthcare systems varies significantly. How will that be taken into account during the economic evaluation?

Response: We understand the concern of reviewer regarding high variations in cost of care across public vs private healthcare systems. We would like to submit, in general there are two cost components for patients seeking care from public healthcare facilities i.e., health system costs and out of pocket expenditures (OOP). While private healthcare facilities being profit-making units charge everything from patients, therefore OOP is a good proxy of cost of care for patients utilising private healthcare. We plan to do cost estimations accordingly. For public sector, the health system cost data will be accessed from open source National Health System Cost Database which provides cost estimates for public health facilities by level of care and speciality. Primary data pertaining to OOP will be collected as a part of study both for a sample of patients covered under clinical trial, both seeking public and private healthcare facilities. We have already provided related text on page 7, table 1; page 11, lines 31-35 and page 12, lines 1-8.

8. Is there a plan to do an a priori subgroup analysis? e.g. based on gestational age?

Response: Accurate measurement of gestational age is difficult to assess by field workers, the study relies on birth weight as its proxy. Sub-group analysis based on gestational age is therefore not planned.

9. How will other co-interventions, e.g. hand hygiene and breastfeeding rates, be compared between the two groups, and is there any plan to adjust for it in the analysis?

Response: There could be a possibility of method of feeding affecting the outcome since formula feeding is associated with greater morbidity. We are collecting data on breastfeeding and if the infant had received any other foods or drinks. We would be able to see whether it makes a difference.

Reviewer-2

Mr. Evans Otieku, University of Ghana, Aarhus University

Comments to the Author:

It is good to attempt trying to publish this protocol for an ongoing economic evaluation study. However, it did not seem that a health economist was part of the protocol design. Some clarity is needed to improve the quality of the protocol.

Thanks for your comments. We would like to clarify that a Health Economists (from PGIMER, Chandigarh, India) are involved in the design of this sub-study on Health economics

1. The study has no clear definition of what neonate means. If it applies to newborns aged <28 days old (<4 weeks) at the time of sampling, how is the 0 to 2 months old fit the definition of neonates?

Response: We understand that the term 'neonate' applies to new-borns aged <28 days

old. The study population is the young infant period including 0-2 months. The newborns will be enrolled in the main study (Sinha et al TRIALS 2021) between 03 to 07 days of their age and will be followed up till 02 months of age. They will be included in this

health economics sub study at the time of study completion (end of study visit) based on their willingness.

2. There is no clear statement of the research question.

Response: We regret for this omission. We have added research question on page 5 above 'objectives' in the revised version of manuscript (page 5, lines 6-8). It read as follows,

'Is probiotic (vivomixx) use as prophylaxis a cost-effective strategy for preventing sepsis among 0-2 month old infant population in India?'

3. The authors intend to use DALY averted as a utility measure. However, DALY weights are predetermined and obtainable from the global burden of disease reports and are not suitable for clinical trial cost-effectiveness analysis. Instead, QALY gained with the intervention is a preferred utility measure.

Response: We thank the reviewer for seeking clarification regarding use of DALY as proposed utility measure in the present economic evaluation. We do agree with reviewer's observation about use of DALY. However, till date there is no EQ-5D version specifically designed for neonates and toddler age group. There are other HRQOL tools which exist for this age group such as Infant Toddler Quality of Life Questionnaire (ITQOL); Toddler and Infant (TANDI)- HRQOL instrument etc. but comparability issues across different age groups exist with use of such tools. Hence, use of DALY is the most appropriate choice and enables to ensure transparent resource allocation among interventions working <4 years age children. Though QALY is a preferred utility measure for economic analysis, but DALY is also recommended in specific circumstances (including LMICs) and widely used for evaluating neonatal interventions 2345 . Lastly, the draft HTA reference case of India also recommends use of DALY in HTAs evaluating neonatal interventions.

4. It is unclear whether the protocol reporting followed specific guidelines. Many sections were misplaced. For example, Figure 1 appeared on page 4 but is referred to on page 9, line 40, as below. It will be best to merge the section on Costing and Estimation of Costs under one heading and include another section on the Measurement of Effectiveness.

2 Wilkinson T, Chalkidou K, Walker D, Lopert R, Teerawattananon Y, Chantarastapornchit V, Santatiwongchai B, Thiboonboon K, Rattanavipapong W, Cairns J, Culyer T.

3 The International Decision Support Initiative (iDSI) Reference Case for Health Economic Evaluation. F1000Research. 2019 Jun 10;8(841):841

4 Ali A, Nudel J, Heberle CR, Olson KR, Hur C. Cost effectiveness of a novel device for improving resuscitation of apneic newborns. BMC pediatrics. 2020 Dec;20(1):1-9

5 Mathewos B, Owen H, Sitrin D, Cousens S, Degefe T, Wall S, Bekele A, Lawn JE, Daviaud E. Community-Based

interventions for newborns in Ethiopia (combine): cost-effectiveness analysis. Health policy and planning. 2017

Oct 1;32(suppl_1):i21-32.

Response: Thanks for pointing out the inconsistency in citing figures in the text. We think the figure 1 (conceptual framework) is relevant to the section 'Theory of Change' on page 3 and 4. We have added additional text under 'Theory of change' and cited figure 1 on page 5, lines 1-2.

We thank the reviewer for highlighting the confusion related to section heading. We would like to clarify; we have given 'Costing' as major heading; and 'sample size' and 'estimation of costs' as sub-headings under 'Costing' section. In the revised version, we use different levels of font for headings and sub-headings. Also, we have carefully proofread the revised manuscript any inconsistency in headings to avoid confusion.

5. Spell out all abbreviations in full at the first instance. For example, EC, ERC, MRC, IWRS, PSBI, AE/SAE, etc.

Response: We have added a list of abbreviations as annexure 1 on page 20.

1. EC: Ethics Committee
2. ERC: It is inadvertently mentioned as ERC in the manuscript; however, it is REC which means Research Ethics Committee (REC) of the Liverpool School of Tropical Health, (LSTM) UK
3. MRC: The Medical Research Council
4. UKRI: United Kingdom Research and Innovation
5. NHSCD: The National Health System Cost Database
6. IWRS: Interactive Web Response Systems
7. PSBI: Possible serious bacterial infection.
8. AE/SAE: Adverse events/ Serious adverse events
9. DALY: Disability adjusted life year
10. LMIC: Low- and Middle-income countries
11. SDG: Sustainable Development Goal
12. LBW: Low birth weight.
13. QALY: The quality-adjusted life-year
14. RCT: Randomized controlled trials
15. CUA: Cost utility analysis
16. SRS: Sample Registration System
17. NHSCD- National Health Systems Costing Database
18. OOP: out-of-pocket
19. TLC: Total Leukocyte Count
20. CRP: C-reactive protein.
21. ESR: Erythrocyte sedimentation rate
22. HBPNC: home based post-natal care
23. NHM: National Health Mission
24. RMNCH: Reproductive, Maternal, New-born, Child, and Adolescent Health
25. CEA: Cost-effectiveness analysis
26. PSA: Probabilistic sensitivity analysis.
27. NHM: National Health Mission
28. ABH: Alcohol-based hand rub
29. BSI: Blood stream infection

30. IECR: Incremental cost-effectiveness ratio
31. CDSCO: Central Drugs Standards Control Organisation
32. IEC: Institutional Ethics Committee

6. Study Trial: Page 8, line 52, presence of any danger sign....." what is this danger sign.

Response: 'Danger sign' is any sign and symptom which needs referral & physician assessment. It can be 'possible serious bacterial infection'/Local bacterial

infection/critical illness etc.

7. Specifically, what is the difference between the primary and secondary outcomes?

Response: Primary outcomes in the main trial include culture proven sepsis/sepsis screen positive and possible serious bacterial infection whereas the secondary outcomes include critical illness, local bacterial infections, evaluation of colonisation of probiotic bacteria in gut at 0, 21 & 60 days and evaluation of cost effectiveness of probiotic intervention. (Sinha et al TRIALS 2021). Conventionally, sample size estimation is done for the primary outcomes, and the same is applicable in our study.

8. How do they intend to characterize heterogeneity and distributional effects if they collect data from six different geographic locations?

Response: The main objective of the study is to evaluate the effect of probiotic intervention in different populations across the country. We have purposively chosen to collect data (individual healthcare systems) from 6-sites participating in the ProSPoNS study and we anticipate a wide variability in reporting individual and institutional-specific cost accounting as an added advantage of the study. Heterogeneity will be assessed by measures of skewness and if required, appropriate distributional transformation will be applied to transform into homogeneous distribution and summarised accordingly. In case, no transformation works out, rank based summary measures will be used. Sitewise distributional effects: Site will be used as a covariate while doing the pooled (all sites combined) analysis.

9. Why didn't they also consider the cost of productivity loss due to presenteeism?

Response: We thank the reviewer for this comment. First, the health condition under consideration is sepsis among the young infant population which is treated in inpatient setting. Therefore, the parents (or/and caregivers) must be present in the hospital during the inpatient stay which clearly indicate absenteeism from work. Second, mostly studies consider cost of productivity losses due to presenteeism mainly for employees suffering with illnesses 67 . Lastly, after hospital discharge, follow up with parents and caregivers to assess any effect on their productivity at work is beyond the scope of data collection

6 Braakman-Jansen LM, Taal E, Kuper IH, van de Laar MA. Productivity loss due to absenteeism and presenteeism by different instruments in patients with RA and subjects without RA. *Rheumatology*. 2012 Feb

1;51(2):354-61

7 Li Y, Zhang J, Wang S, Guo S. The effect of presenteeism on productivity loss in nurses: the mediation of

health and the moderation of general self-efficacy. *Frontiers in psychology*. 2019 Jul 31;10:1745.

in the present study. Additional text is added in revised version of manuscript to state this part clearly (page 10, lines 32-33 and page 11, lines 1-2).

10. Intext citation styles are not consistent. Some come after a period. Some are enclosed in (), while others are in [], making it difficult to read as values or references.

Response: We thank the reviewer for the comment. We have now thoroughly checked and made necessary corrections in the modified version of the manuscript.

11. The authors said all neonates will be followed up at home for the first seven days. Are they not anticipating some neonates will be at the hospital due to their low birth weight? Does it mean if a neonate is still on admission they will be excluded? What exactly is their exclusion criteria?

Response: The healthy low birth weight infants, if eligible for inclusion in the study will be enrolled between day 3-7 of life at the hospital/facility and followed up after discharge, at home till they are 60 days old. During follow-up they would be referred to study physician at the hospital in case of any illness and be hospitalized if required.

As a routine practice in India, healthy low birth weight infants are discharged from the hospital after delivery if they are not sick. Only extremely LBW /premature babies in need of NICU treatment are retained in the hospital after birth. Such babies are not being enrolled in the present trial.

Therefore, to answer the reviewer's query if a neonate is still on admission they will be excluded, is correct. We understand that some neonates will be in hospital due to their low birth weight; if a neonate is on admission he will be followed up in the hospital and not enrolled.

The exclusion criteria as per the protocol are:

1. An infant with extreme prematurity (< 34 weeks)
2. An infant with illness requiring prolonged hospitalization and interference with oral feeds
3. Presence of a gross congenital malformation incompatible with life
4. Parent/legal authorized representative (LAR) not providing written consent

12. They should rephrase the following for clarity:

Page 3, lines 19 to 25, and lines 35 to 39. They should delete lines 46 to 48 (repetition).

Response: We thank the reviewer for this comment. We have now rephrased lines 19-25 to state that 'The primary objective of the health economics study is to assess the cost –utility of using probiotics as prophylaxis for sepsis among low birth weight neonates of age 0-2 months'. Lines 35-39 refer to earlier study by Panigrahi et al reporting incidence of sepsis. Lines 46-48 state the increased mortality among LBW infants. Therefore there is no repetition.

Reviewer: 3

Dr. Jayashree Purkayastha, Kasturba Medical College Hospital

Comments to the Author:

Whether the babies included in the study are preterm or term, SGA, AGA is not mentioned. Only LBW is mentioned which has to be clearly defined. if preterm babies included then they will have prolonged NICU stay, which day of life you will start the probiotic is not defined. PSBI and critical illness are overlapping. Why did you choose Vivomixx as the probiotic in the study should be clarified.

Response: The inclusion criteria are listed below:

1. Infant birth weight ≥ 1500 to ≤ 2500 g.
2. Age of the infant on recruitment falls between 3rd and 7th day of life,
3. Stable clinical condition as assessed by the physician ("stable" means one who does not require intravenous fluids and vasopressor medication to maintain circulation and accepts oral feeding/ breastfeeding).
4. The mother (with the new-born) is planning to stay in the study area for a period of at least two months

Assessment of gestational weight is not included.

1. Please mention the cut off gestational age and birth weight

Response: The cut off for gestational age: ≤ 34 weeks

2. Birth weight:

Response: Birth weight ≥ 1500 g to < 2500 g

3. Which day of life you will start the probiotic:

Response: Probiotics will be started between day 03 to day 07 of life of the participant.

4. PSBI and critical illness are overlapping

Response: We agree with the reviewer. As per the definition given by the WHO/UNICEF, there is some overlap in 'movement' and 'feeding' related questions, we understand that although there is overlap, the wordings indicate a grading of severity in clinical condition. 'unable to feed' is in Critical illness, whereas 'not feeding well' in PSBI. Similarly,

'Movement only on stimulation' or 'No movement at all' is included in PSBI and 'No movement on stimulation' is in critical illness.

5. Why did you choose Vivomixx as the probiotic in the study should be clarified:

Response: The interventional product Vivomixx is a combination of 08 probiotics with proven efficacy in NEC. The interventional product was used in the Pilot study conducted by ICMR for prevention of neonatal sepsis. This study will complement the already completed ICMR study by producing conclusive evidence regarding the efficacy and safety of the probiotic blend in low birth weight infants.

VERSION 2 – REVIEW

REVIEWER	Singh, Balpreet Dalhousie University
REVIEW RETURNED	14-Feb-2023

GENERAL COMMENTS	Thank you for providing detailed clarifications. I have one follow-up comment. 1. Why are there two primary outcomes - sepsis and possible serious bacterial infections? Why not just select sepsis as a primary outcome?
--

REVIEWER	Otieku, Evans University of Ghana, Institute of Statistical, Social and Economic Research
REVIEW RETURNED	03-Feb-2023

GENERAL COMMENTS	The author's response to my initial review comments is satisfactory and approved for publication.
---

REVIEWER	Purkayastha, Jayashree Kasturba Medical College Hospital, Paediatrics
REVIEW RETURNED	27-Jan-2023

GENERAL COMMENTS	Authors have done the corrections as recommended by the reviewers
---

VERSION 2 – AUTHOR RESPONSE

Reviewer: 1

Dr. Balpreet Singh, Dalhousie University

1. Why are there two primary outcomes - sepsis and possible serious bacterial infections? Why not just select sepsis as a primary outcome?

Response: Culture positivity rates are very low and vary between different labs. The sample size calculations were not done for this outcome. Physician diagnosed sepsis in PSBI our earlier study (BMJ OPEN 2015) provided the data for sample size calculations.

VERSION 3 – REVIEW

REVIEWER	Singh, Balpreet Dalhousie University
REVIEW RETURNED	01-Mar-2023
GENERAL COMMENTS	No further comments.